# New Antimicrobial Biomaterials for the Reconstruction of Craniofacial Bone Defects

Andreea Elena Miron (Lungu) [1], Marioara Moldovan [2] 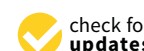, Cristina Alexandra Prejmerean [2], Doina Prodan [2,*], Mihaela Vlassa [2], Miuța Filip [2], Mîndra Eugenia Badea [1] and Mădălina Anca Moldovan [3]

[1] Department of Prevention in Dental Medicine, University of Medicine and Farmacy Iuliu Hațieganu, Dental Medicine, 31 Avram Iancu Str., 400083 Cluj-Napoca, Romania; miron.andreeaelena@gmail.com (A.E.M.); mindrabadea@gmail.com (M.E.B.)

[2] Department of Polymer Composites, Raluca Ripan Institute for Research in Chemistry, Babeș-Bolyai University, 30 Fântânele Str., 400294 Cluj-Napoca, Romania; marioara.moldovan@ubbcluj.ro (M.M.); cristina.alexandra.febr.2016@gmail.com (C.A.P.); mihaela.vlassa@ubbcluj.ro (M.V.); filip.miuta@ubbcluj.ro (M.F.)

[3] Department of Cranio-Maxillo-Facial Surgery, University of Medicine and Farmacy Iuliu Hațieganu, Dental Medicine, 33 Calea Moților Str., 400001 Cluj-Napoca, Romania; madalina.lazar@umfcluj.ro

\* Correspondence: doina_prodan@yahoo.com; Tel.: +40-724-254-336

**Abstract:** Reconstructive bone surgery of the head and neck could prove challenging in terms of postoperative healing and recovery. Fighting infection during the healing period is one of the critical factors of the long-term survival of an implant. The aim of the study was to develop an innovative composition suitable for an antibacterial craniofacial implant that should have the capacity to continuously and constantly release the amount of gentamicin necessary to prevent the post-surgical infections. For this purpose, a series of composite materials based on dimethacrylic monomers, hydroxyapatite and $ZrO_2$, with (series B) or without the addition of polymethyl methacrylate (series A), reinforced with woven E-glass fibers (FRC) were obtained using the laminate lay-up process. Gentamicin was included in all FRC sample matrices to confer an antimicrobial effect. The results show that after extraction of the residual monomers from the FRC samples in different solvents (chloroform, acetone and ethyl alcohol), the cumulative amount of released gentamicin after 12 days was between 7.05–11.38 mg for A samples and 11.21–14.52 mg for B samples. The microbiological protocol showed that gentamicin induces a two weeks-lasting antimicrobial effect maintained over the minimal inhibitory concentration for *P. aeruginosa* and *S. aureus*.

**Keywords:** polymer; fiberglass; gentamicin release; residual monomers; antimicrobial activity

## 1. Introduction

Despite the development that the microsurgical and tissue engineering techniques have undergone, alloplastic materials are still widely used in craniofacial reconstructive surgery [1–3]. Being a cheap and available on-the-market alternative, the advantages of these materials include resistance to functional and external forces, secure handling during surgery, insignificant resorption rate, and favorable cosmetic results due to the possibility of prefabrication of customized implants using additive manufacturing techniques [4].

The requirements that need to be met for the host positive response and function after the defect is grafted can be very demanding. One of the most important issues of composite materials is the release of residual monomer during polymerization and afterwards. [5,6]. Giving that the residual monomer

is known to induce cytotoxicity and genotoxicity, as well as potential allergic response, researchers struggle to decrease residual monomer release [7].

Another problem related to alloplastic materials used for bone reconstruction is the infection. In the case of using alloplastic implants based on titanium alloys or synthetic polymers such as porous polyethylene and polymethyl methacrylate (PMMA), infection rates vary from 3% to 40%. The presence of signs and symptoms of infectious disease at the implantation site may urge for implant removal [8–10]. Although the implants are sterilized, the surgical procedures comply with the rules of asepsis and antisepsis, antibiotics are administered systemically before and after the surgery, and the rates of infection are not significantly reduced [11]. As a result, current research focuses on local antimicrobial strategies to prevent implant-associated infections and to reduce the number of complications and surgical re-interventions. A first prophylactic strategy refers to the "race for the surface", a concept described 30 years ago by the orthopedist A. G. Gristina [12]. The fate of alloplastic implants is considered to be a "race" between bacterial adhesion, with the formation of the biofilm on the surface of the implant and its tissue integration, ensured by the growth of soft tissues and bone on or within the alloplastic material [13–15]. In order to have antibacterial behavior, the implant surface should limit the formation of the biofilm, fulfilling the following objectives: to inhibit bacterial adhesion, to have bactericidal action and to exhibit antimicrobial effect after implantation. In general, two of these strategies are combined to overcome the limitations of using a single property [16–18]. Gentamicin is a broad-spectrum antibiotic [19], usually administered intravenously, intramuscularly or locally to treat bacterial infections. The efficiency of gentamicin in the treatment of maxillofacial and especially bone infections has been proven by numerous clinical studies [20–22].

Joining the constant preoccupations in the field, namely, to obtain an implant with improved performances to be used for the reconstruction of craniofacial bone defects, we have previously studied fiber-reinforced composites based on polymeric matrices reinforced with woven E-glass fibers. The polymeric matrices were obtained from dimethacrylic monomers: 2,2-bis-[4-(2-hydroxy-3-methacryloyloxypropoxy)phenyl]-propane) (Bis-GMA), triethylene glycol dimethacrylate (TEGDMA), urethane-dimethacrylate (UDMA), hydroxyethylmethacrylate (HEMA) and woven E-glass fibers (300 g/mp) that were previously treated with methacryloxypropyl trimethoxysilane coupling agent (A-174) to ensure good interfacial adhesion and stress transfer across the interface. The materials developed achieved both biocompatibility and mechanical criteria requested for craniofacial bone reconstruction [6,22].

The main objective of the present study was to obtain a series of composite materials reinforced with woven E-glass fibers (FRCs) by the laminate lay-up process to be used as antimicrobial craniofacial custom implants. For this purpose, gentamicin was included in the matrix (filled dimethacrylate resins with and without addition of PMMA) of the FRCs to prolong protection against infections as long as it is needed for the implant to integrate without administering a high dosage of systemic antibiotic.

The novelty of this study is represented by the composition of FRCs, mainly by the fact that the gentamicin was introduced as a component in the composition of the FRC matrices, not just in the outer layer of the material. The scientific hypothesis to be tested is that the incorporation of gentamicin in the whole mass of the FRC matrix derived from semi-interpenetrating polymer network will lead to the obtaining of an implant that will have the capacity to continuously and constantly release the required amount of gentamicin in order to prevent postoperative infections.

## 2. Materials and Methods

*2.1. Experimental Materials*

2.1.1. Materials

Triethylene glycol dimethacrylate (TEGDMA), benzoyl peroxide (POB), N, N-Bis(2-hydroxyethyl)-p-toluidine (DHEPT), butylated hydroxytoluene (BHT) and $ZrO_2$ were purchased from Sigma-Aldrich Chemical Co. (Taufkirchen, Germany) and used without additional

purification. Bis-GMA: analogue (93% 2,2-Bis[p-(2-hydroxy-3-methacryloyloxypropoxy)-phenyl]-propane monomer and 7% dimer) was obtained at the Babeș-Bolyai University, Raluca Ripan Institute for Research in Chemistry, (Cluj-Napoca, Romania) [23]. Hydroxyapatite (HA) was also synthesized in the laboratory of UBB-ICCRR [24]. Methyl methacrylate (MMA) and polymethyl methacrylate (PMMA) was purchased from Spofa Dental, Czech Republic. Gentamycin sulfate BioChemica was purchased from AppliChem Panreac., Miami, FL, USA. The woven E-glass fibers were purchased from Owens Corning, Brussels, Belgium.

### 2.1.2. Preparation of FRC Samples

#### The Matrices

Two matrices (A and B) for the obtaining of FRC samples were prepared in the form of filled resins with chemical initiation of polymerization. Two filled resins, one containing the catalyst (1% by weight benzoyl peroxide (POB)) and one containing the accelerator of polymerization (1% N, N-dihydroxy ethyl-p-toluidine (DHEPT) were prepared for each matrix. The two matrices (A and B) contained the same filler, namely hydroxyapatite (as a regulator of viscosity and as a nano-filler with increased bioactivity and biocompatibility), zinc oxide (radio-opacifying agent) and gentamicin for antimicrobial effect. After the mixing of the three components, the filler was uniformly dispersed in the resins, two containing Bis-GMA and TEGDMA as monomer mixtures (for matrix A), and two containing Bis-GMA, TEGDMA and PMMA (for matrix B), in one of each mixture having different composition being previously dissolved the POB catalyst, and in the others, the DHEPT accelerator of polymerization. The FRCs filled resins composition is shown in Table 1.

**Table 1.** FRC matrices composition, the thermic treatment and the solvent used to extract the residual monomer.

| Samples Code | Bis-GMA (%) | TEGDMA (%) | PMMA (%) | HA (%) | ZrO$_2$ (%) | Gentamicin (%) | Thermic Treatment | Solvent |
|---|---|---|---|---|---|---|---|---|
| 1A | 35.64 | 23.76 | - | 6.6 | 12 | 22 | - | chloroform |
| 2A | 35.64 | 23.76 | - | 6.6 | 12 | 22 | 100 °C, 2 h | chloroform |
| 3A | 35.64 | 23.76 | - | 6.6 | 12 | 22 | - | ethyl alcohol |
| 4A | 35.64 | 23.76 | - | 6.6 | 12 | 22 | - | acetone |
| 1B | 17.82 | 23.76 | 17.82 | 6.6 | 12 | 22 | - | chloroform |
| 2B | 17.82 | 23.76 | 17.82 | 6.6 | 12 | 22 | 100 °C, 2 h | chloroform |
| 3B | 17.82 | 23.76 | 17.82 | 6.6 | 12 | 22 | - | ethyl alcohol |
| 4B | 17.82 | 23.76 | 17.82 | 6.6 | 12 | 22 | - | acetone |

#### Silanization of the Woven E-Glass Fibers

E-glass fibers were treated with a 1 wt.% A-174 (γ methacryloxypropyl-1-trimethoxysilane) coupling agent. The woven E-glass were immersed for 1 h in the silane solution (A-174 silane in ethanol–water 90/10 vol.%) and the pH level was 3.8 maintained using glacial acetic acid; the silane layer was dried at 110 °C for 2 h.

#### The Cured FRCs

Eight FRCs used for gentamicin release; residual monomers and scanning electron microscopy (SEM) analysis were obtained into Teflon molds with a diameter of 15 mm and a height of 1 mm. The matrices were reinforced with the woven fiber glass by following a laminate lay-up process [2,22]. The sequence of the layers was: the matrix containing the accelerator, the woven fiberglass treated with silane A-174 and the matrix containing the catalyst. A glass lid was placed on the free surface of the mold, and on top of it, a weight of 100 g was placed to remove the excessive resin by compressing the specimen. The final content of the glass fiber in the FRC was 63%. The final curing time was 30 min. An additional thermic treatment was applied after initial polymerization (shown for two FRC

samples). Post-FRC polymerization was performed in the electric oven at 100 degrees Celsius for 2 h. After cooling, the samples were kept drying for 2 h in a standard dryer. After drying, each sample was weighed on the analytical scale and stored in a labeled box to avoid confusing the samples.

*2.2. Caracterisation of Experimental Materials*

2.2.1. Residual Monomers

The cured FRC samples, weighed previously, were immersed in 10 mL of solvent for 8 h at room temperature. Three types of solvents were used: acetone, ethyl alcohol and chloroform in order to evaluate the removal of the residual monomer. The samples were extracted from the solvent with tweezers, buffered with absorbent paper and kept in the dryer until constant weight. The extracts were evaporated in vacuo to determine the unreacted monomer from each FRC, and the residue was dissolved in 2 mL acetonitrile. The protocol was described elsewhere [2,5,6].

The analyses were carried out on a Jasco Chromatograph (Jasco International Co., Ltd., Tokyo, Japan) that was equipped with an intelligent PU-980 pump, a LG-980-02 ternary gradient unit, an CO-2060 Plus intelligent column thermostat, an intelligent UV-975 detector and an injection valve that was equipped with a 20 μL sample loop (Rheodyne, Thermo Fischer Scientific, Waltham, MA, USA). The system was controlled and the experimental data were analyzed with the ChromPass software (version v1.7, Jasco International Co., Ltd., Tokyo, Japan). Separation was carried out on a Lichrosorb RP-C18 column (25 cm × 0.46 cm) at a 21 °C column temperature. The mobile phase was a mixture of acetonitrile (A, HPLC grade) and water (B, Milipore ultrapure water), and a gradient was applied according to the following method: 0–15 min. linear gradient 50%–80% A; 15–25 min. linear gradient 80%–50% A. The flow rate was 0.9 mL/min. UV detection was performed at 204 nm for monitoring the elution of BisGMA and TEGDMA analytes. Stock solutions of reference standards of Bis-GMA and TEGDMA (1 mg/mL) were prepared in acetonitrile and were stored at 4 °C. The linearity of the response to analytes was established with four concentration levels and the regression factors $R2$ were higher than 0.998. Three parallel determinations of the residual monomer from each FRC composition were performed.

The residual monomer amount has been determined from the HPLC chromatograms of the extracts and it was calculated as percent related to the initial amount of the monomer in the sample and related to the weight of the sample, respectively.

Table 1 represents the FRC paste composition, the thermic treatment and the solvent used to extract the residual monomer.

2.2.2. Cumulative Gentamicin Release

Gentamicin is a complex mixture of several substances with similar structures [25]. Of these structures, the most important are gentamicin C1, C1a, C2, C2a and C2b. The chemical structure of these types of gentamicin is shown in Figure 1. For quantification of the gentamicin released from the samples, a quantitative liquid chromatographic method (HPLC) with pre-column derivatization using phenyl isocyanate (PIC) as the derivatizing agent was used. After the residual monomer extraction, each sample was reweighed and immersed in a 1.5 mL phosphate-buffered saline solution (PBS), solution maintained in a thermostatic water bath at 37 °C for 24 h. The method was first described by Kim et al. [26]. By using PBS instead of water, we have mimicked the in vivo conditions.

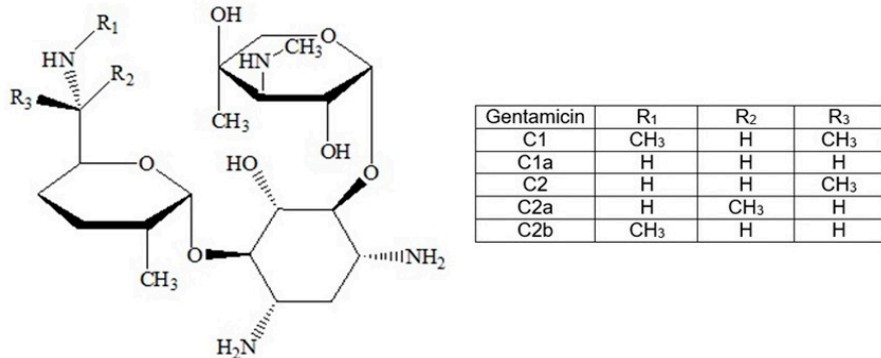

| Gentamicin | R₁ | R₂ | R₃ |
|---|---|---|---|
| C1 | CH₃ | H | CH₃ |
| C1a | H | H | H |
| C2 | H | H | CH₃ |
| C2a | H | CH₃ | H |
| C2b | CH₃ | H | H |

**Figure 1.** Chemical structure of the most important types of gentamicin.

Extraction solutions were collected every 24 h for 14 days. The collected solutions were kept in the freezer (−20 °C) until their HPLC analysis in order to determine the amount of gentamicin released.

The samples were analyzed on a HPLC Agilent 1200 series chromatograph (Agilent Technologies, Morge, Switzerland), equipped with G1311A quaternary pump, G1322A degasser, G1315D DAD detector, G1329A autosampler and G1316B TCC SL column thermostat. ChemStation software (version B.04.01, Waldbronn, Germany) was used to collect and process the chromatographic data. Separation was carried out on a chromatographic column Phenomenex Gemini, 5 μm C18 110A, (250 mm × 4.6 mm) at 25 °C. The mobile phase was a mixture of acetonitrile: water (50:50 *v/v*), acetonitrile for chromatographic use (Optigrade Promochem, Germany) and water (Millipore MilliQ Simplicity ultra-pure water obtainer). The mobile phase is filtered and degassed through a 0.45 μm membrane filter (Teknokroma nylon membrane disc filters.). The flow rate was 1.5 mL/min and the injector volume was 20 μL. Diode-array detection (DAD) was performed at 240 nm for monitoring the elution of gentamicin derivatives. The gentamicin (Gentamycin sulfate BioChemica, activity 641 I.U./mg; lot: 3Q008363) standard solution of 1.0 mg/mL was prepared PBS (PBS of 7.4 pH was prepared with the following chemical composition: $Na_2HPO_4·7H_2O$ (20.214 g/L), $NaH_2PO_4·H_2O$ (3.394 g/L). 250 μL phenylisocyanate (PIC, Fluka, Germany) and 250 μL triethylamine (TEA, Fluka, Germany) were added to an aliquot of 500 μL of gentamicin solution. The PIC and TEA concentrations were 5 μg/mL each in acetonitrile (ACN).

The derivatization reaction takes place at room temperature, the mixtures being shaken few times. The calibrating curves were constructed on domain of concentration: 500–3125 μg/mL. There are five peaks of derivatized gentamicin C1a, C2a, C2b, C1 and C2. Figure 2 shows HPLC chromatograms of 500 μg/mL gentamicin standard solution and a gentamicin sample extract. For all derivatives, the correlation coefficient was >0.9998. The limits of detection (LOD), were between 1–1.5 μg/mL and the limits of quantification (LOQ) were 3.2–4.5 μg/mL, respectively.

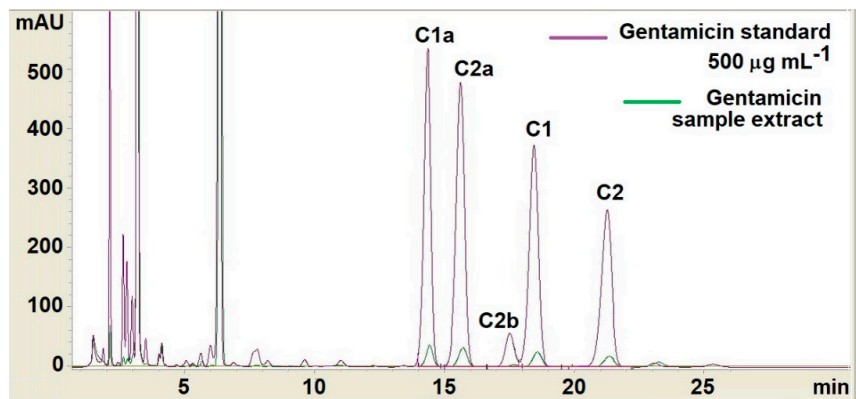

**Figure 2.** Overlapped chromatograms of a derivatized solution of 500 μg/mL gentamicin standard solution and a gentamicin sample extract.

The amount of antibiotic released was assayed by comparison with the calibration curves for the gentamicin derivatives (C1a, C2a, C2b, C1 and C2) made in PBS. The samples were done in triplicate.

### 2.2.3. SEM Investigation of FRC Surface

After storing the samples for 8 h in different solvents and maintaining them for 12 days in PBS to determine the amount of gentamicin released, the sample surfaces were investigated by scanning electron microscopy (SEM) using an INSPECT S (FEI Company, Hillsboro, OR, USA) electron microscope in low vacuum mode.

### 2.2.4. Antimicrobial Activity in Dynamics

The study protocol was applied to determine how gentamicin release from the surface of composite implants is performed over a given period. Furthermore, our research aimed at discovering to what extent the released gentamicin maintains its local concentration over MIC (minimum inhibitory concentration) and MBC (minimum bactericidal concentration) for the two different bacterial strains (*Staphylococcus aureus* ATCC-25923 and *Pseudomonas aeruginosa* ATCC-27853). The tests were performed in duplicate.

Two bacterial strains were used for biological tests: *Pseudomonas aeruginosa* (ATCC 27853) and *Staphylococcus aureus* (ATCC 25923). The microorganisms tested were obtained from the Research Center in Agro-Food Biochemistry and Biotechnology of the Institute of Life Sciences of the University of Agricultural Sciences and Veterinary Medicine, Cluj-Napoca, Romania. Bacteria were grown on Mueller-Hinton agar (BioMérieux, Marcy l'Etoile, France), and the cultures were stored at 4 °C and subcultured once a month. Prior to starting each test, bacteria were grown overnight in 5 mL of Mueller-Hinton broth (BioMérieux, Marcy l'Etoile, France) in a shaker incubator (Heidolph Inkubator 1000 coupled with HeidolphUnimax 1010, Schawbach, Germany) at 37 °C, 150 rpm until the culture is formed.

The inoculation ring, previously sterilized by flaming, was soaked in Mueller-Hinton broth. Subsequently, with the same inoculation loop, striations were made on a Petri plate with Mueller-Hinton agar culture medium using the three-phase striatum model known as T-Streack. The technique was used to isolate a pure strain from a single species of microorganism. Petri dishes containing culture media were then incubated at 37 °C for 18 h.

Enough colonies were collected, which were placed in 9 mL of sterile saline. The concentration of bacteria corresponding to $10^7$ colony-forming units (CFU) per mL ($10^7$ CFU/mL) was determined using the ND-1000 NanoDrop Spectrophotometer (Delaware City, DE, USA). Successive dilutions up to $10^5$ CFU/mL were obtained.

Determination of MIC (Minimum Inhibitory Concentration) and MBC (Minimum Bactericidal Concentration)

A modified variant of the microdilution technique was used to evaluate antimicrobial activity [27,28].

Bacterial strains (*Staphyloccocus aureus* ATCC-25923 and *Pseudomonas aeruginosa* ATCC-27853) were grown overnight at 37 °C on tryptic soy broth (TSB). The bacterial suspensions obtained were diluted in physiological serum until a concentration of $2 \times 10^5$ CFU/mL was obtained.

Dilutions from the obtained inoculum were cultured on Mueller-Hinton (MH) solid medium to certify the absence of bacterial contamination and to validate the obtained inoculum. The determination of the minimum inhibitory concentrations (MICs) was performed by the successive dilution technique, using 96-well microtiter plates.

Different dilutions of FRC-G extracts obtained in 100 µL of Mueller-Hinton broth (MH) were introduced into the wells; then, 10 µL of inoculum was added to each of the 96 wells of each microtiter plate. The plates were incubated at 37 °C for 24–48 h.

The minimum inhibitory concentrations of the obtained extracts were determined after the addition of 20 µL (0.2 mg/mL) of resazurine solution in each of the wells, followed by incubation of the plates for another two hours at 37 °C.

The change of color, from blue to pink, indicated the reduction of the resazurine solution, and thus the bacterial growth. The minimum inhibitory concentration was defined as the lowest concentration of the extract that prevented the color change. The minimum bactericidal concentration was determined by subcultivation of 2 µL of the resulting suspensions.

The sample size which was used for evaluating the antibacterial activity of the FRC discs was 6 mm × 2 mm.

## 3. Results

### 3.1. Residual Monomers

Figure 3 shows the HPLC chromatograms of the FRC sample extracts investigated:

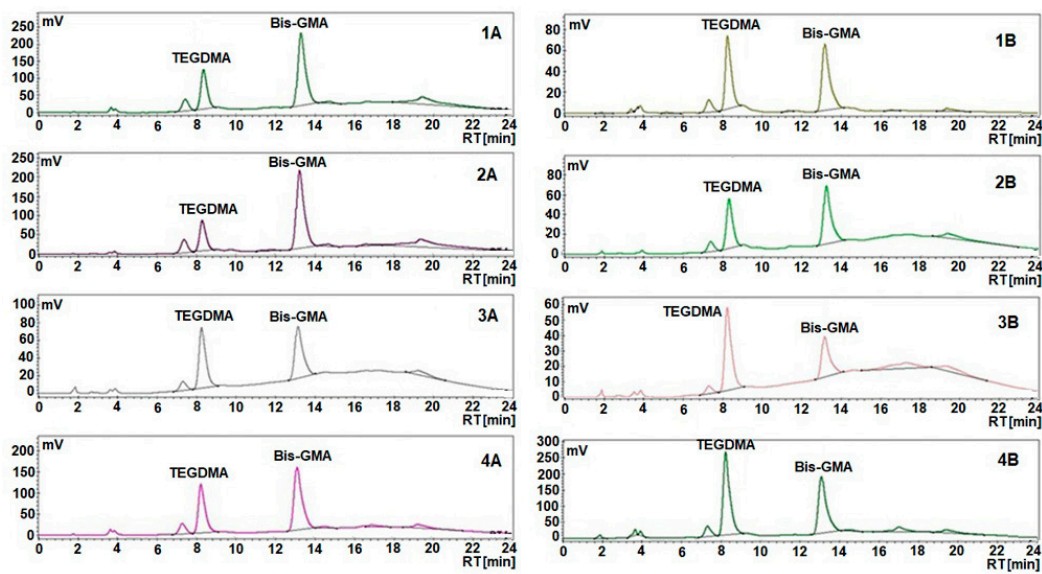

**Figure 3.** HPLC chromatogram of residual monomers from the FRC extracts samples.

Based on the calibration curves and the presented HPLC chromatograms, the amounts of monomers extracted from the hardened FRC composites were determined. The percentages of extracted Bis-GMA and TEGDMA monomers relative to the corresponding monomer quantities introduced into the initial monomer mixtures are shown in Table 2:

**Table 2.** The percentages of extracted Bis-GMA and TEGDMA monomers.

| Sample Code | Initial Bis-GMA (g) | Extracted Bis-GMA (mg/2 mL) | Extracted Bis-GMA (%) | Initial TEGDMA (g) | Extracted TEGDMA (mg/2 mL) | Extracted TEGDMA (%) |
|---|---|---|---|---|---|---|
| 1A | 0.068 | 1.168 | 1.7 | 0.045 | 0.709 | 1.6 |
| 2A | 0.068 | 1.120 | 1.6 | 0.045 | 0.466 | 1 |
| 3A | 0.058 | 0.258 | 0.4 | 0.039 | 0.431 | 1.1 |
| 4A | 0.062 | 0.812 | 1.3 | 0.041 | 0.753 | 1.8 |
| 1B | 0.030 | 0.363 | 1.2 | 0.030 | 0.430 | 1.4 |
| 2B | 0.030 | 0.240 | 0.8 | 0.030 | 0.276 | 0.9 |
| 3B | 0.028 | 0.056 | 0.2 | 0.028 | 0.325 | 1.2 |
| 4B | 0.039 | 0.985 | 2.5 | 0.039 | 1.724 | 4.4 |

Table 2 shows comparable percentages of extracted residual monomers for samples A (1A–4A) while using different solvents. The percentage of Bis-GMA extracted is between 0.4% when using ethyl alcohol as the extraction solvent and 1.7% when using chloroform. As for the TEGDMA monomer, it is extracted in the highest quantity (1.8%) in the case of acetone and the smallest amount when using ethyl alcohol (1%). There are slight differences between samples 1A and 2A, which means that heat treatment does not decisively influence the amount of residual monomer, especially in the case of Bis-GMA monomer (1.7% Bis-GMA extracted from sample 1A against 1.6% extracted from sample 2A and 1.6%, respectively; TEGDMA extracted from sample 1A versus 1% TEGDMA extracted from sample 2A).

It can also be noted that the highest percentages of the extracted monomers are in the case of using chloroform and acetone, respectively, and that these percentages are close in value (1.7% Bis-GMA extracted from sample 1A as compared to 1.3% Bis-GMA extracted from sample 4A and 1.6% TEGDMA extracted from sample 1A comparable to 1.8% extracted from sample 4A). This means that the two solvents, chloroform and acetone, are the best solvents that can be used for the evaluation of the total extraction of residual monomers from FRC composites.

The same behavior can be observed in the case of samples B (1B–4B); however, it is noted that in the case of sample 4B, the extracted residual monomer is twice as high as in the case of sample 4A.

### 3.2. Cumulative Gentamicin Release

Figures 4 and 5 present the cumulative amount of gentamicin released within 12 days from the investigated FRC samples.

Analyzing the cumulative amount of gentamicin released within 12 days from the investigated FRC samples (Figure 4), it can be seen that the most significant amount of gentamicin is released in the case of sample 4A stored in acetone, followed by sample 3A immersed in ethyl alcohol. Samples maintained in chloroform 1A and 2A release about 2/3 of the amount of gentamicin released by sample 4A. Of the samples 1A and 2A, the heat-treated sample 2A releases a smaller amount of gentamicin, which leads to the idea that by heat treatment, as the conversion increases, the crosslinking density also increases, which causes a smaller amount of gentamicin to be released in the same period of time.

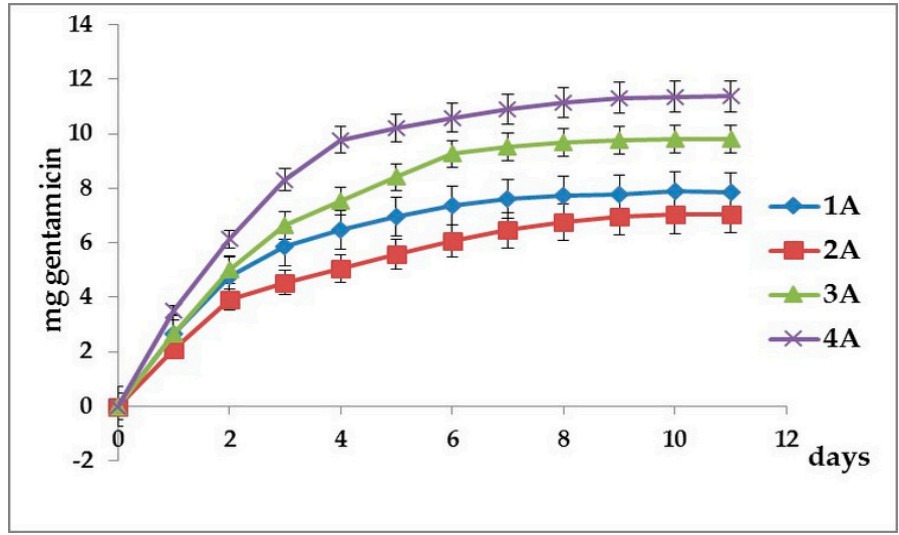

**Figure 4.** Cumulative amount of gentamicin released over time from samples 1A–4A.

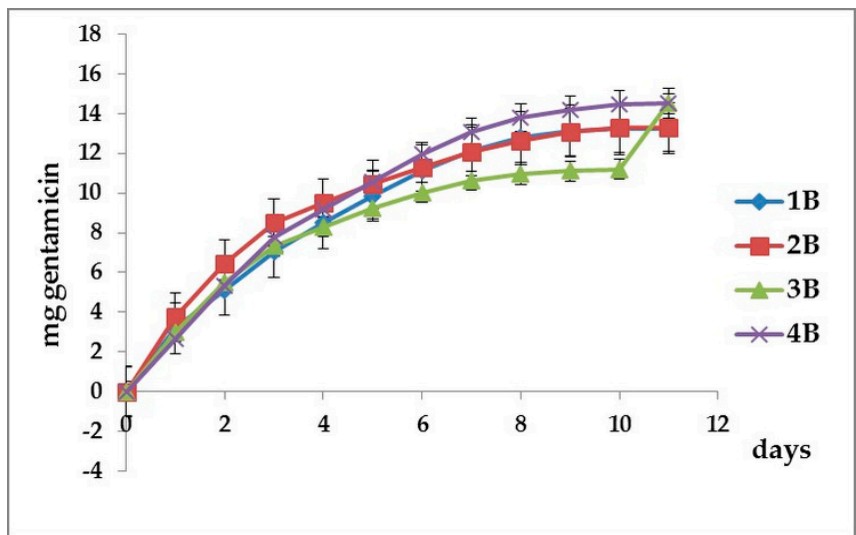

**Figure 5.** The cumulative amount of gentamicin released over time from samples 1B–4B.

It can also be seen that in the first 7 days, a higher amount of gentamicin was released and in the following days, the amount of released gentamicin was 1 or even 2 orders of magnitude smaller. After 12 days, the cumulative amount of released gentamicin was between 7.05 mg for sample 2A and 11.38 mg for sample 4A.

In the case of samples contained in the organic matrix, the PMMA polymer in addition to Bis-GMA and TEGDMA (Figure 5), it can be seen that for the samples stored in chloroform and acetone, respectively (1B, 2B and 4B), the amount of gentamicin released is higher than for the corresponding samples containing only Bis-GMA and TEGDMA (1A, 2A and 4A). This behavior can be explained by the fact that the substitution of an amount of Bis-GMA in the matrix with the linear polymer PMMA leads to the formation of a semi-interpenetrating matrix with a lower crosslinking density, from which a larger amount of gentamicin can be released. However, in the case of ethyl alcohol use, about the same amount of gentamicin is released for both organic matrices studied until the tenth day; following that day, the amount of gentamicin released will increase sharply, reaching day 12 to the value for the sample 4B, maintained in acetone. After 12 days, the cumulative amount of released gentamicin was between 11.21 mg for sample 3B and 14.52 mg for sample 4B.

### 3.3. SEM Investigation of FRC Surface

The investigation of the surface morphology of FRCs after residual monomer extraction and gentamicin release evaluation was performed by scanning electron microscopy (SEM).

From the SEM micrographs (Figure 6), it can be seen that after immersion in chloroform, the texture of the sample surface is substantially modified by the appearance of signs of swollen surface, macropores visible to the free eye and cracks in the polymeric matrix. In the case of using ethyl alcohol or acetone as an extraction solvent, the surface texture of the samples remains relatively smooth without cracks and showing pores, most with the size of about 1 micron, next to a few larger pores with the size of 5–10 μm.

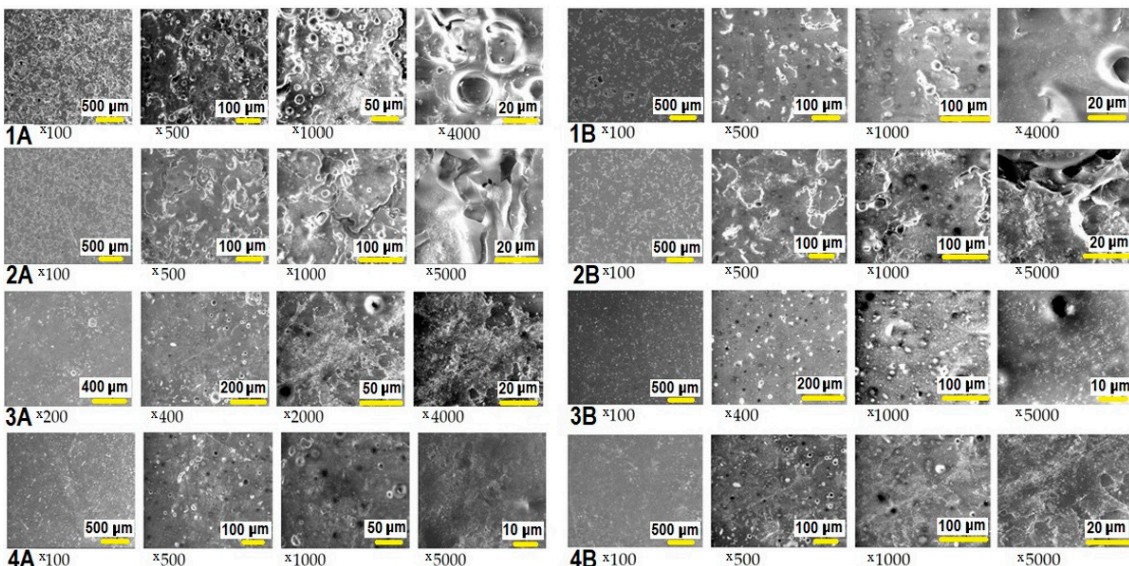

**Figure 6.** SEM micrographs of the FRC samples after extraction and gentamicin release.

### 3.4. Antimicrobial Activity in Dynamics. MIC and MBC Determination

Tables 3 and 4 describe the daily determination of MIC and MBC for the extracts obtained from A and B pills.

The values obtained for the MIC for both *Staphylococcus aureus* ATCC-25923 (0.151 ± 0.019 µg/mL and 0.156 ± 0.033 µg/mL, respectively) and for *Pseudomonas aeruginosa* ATCC-27853 (0.652 ± 0.088 µg/mL and 0.646 ± 0.158 µg/mL, respectively) falls between the reference values in the literature (0.12–1 µg/mL for *Staphylococcus aureus* and 0.5–2 µg/mL, respectively, for *Pseudomonas aeruginosa*). Further studies should focus on optimizing the diffusion of gentamicin from custom implants. Specifically, the local concentration it achieves must be kept within therapeutic limits (5–10 µL/mL) at least one to two weeks after implantation of the biomaterial in order to maximize the efficiency of the method. Furthermore, the local gentamicin concentration should not reach toxic levels (over 10 µL/mL), which could affect the viability of the bone cells in the implant bed.

**Table 3.** Daily determination of MIC and MBC for the extracts obtained from pill A.

| Day | Released Quantity of Gentamicin (µg/mL) | MIC (µg/mL) | | MBC (µg/mL) | |
|---|---|---|---|---|---|
| | | *S. aureus* | *P. aeruginosa* | *S. aureus* | *P. aeruginosa* |
| 1 | 160.00 | 0.15 | 0.62 | 0.30 | 1.24 |
| 2 | 22.00 | 0.17 | 0.68 | 0.34 | 1.36 |
| 3 | 11.02 | 0.16 | 0.68 | 0.32 | 1.36 |
| 4 | 9.66 | 0.15 | 0.6 | 0.30 | 1.2 |
| 5 | 5.60 | 0.16 | 0.7 | 0.32 | 1.4 |
| 6 | 5.42 | 0.16 | 0.66 | 0.32 | 1.32 |
| 7 | 5.02 | 0.15 | 0.62 | 0.30 | 1.24 |
| 8 | 4.90 | 0.15 | 0.6 | 0.30 | 1.2 |
| 9 | 3.58 | 0.11 | 0.88 | 0.22 | 1.76 |
| 10 | 4.42 | 0.13 | 0.55 | 0.26 | 1.1 |
| 11 | 4.00 | 0.12 | 0.5 | 0.24 | 1 |
| 12 | 3.88 | 0.18 | 0.72 | 0.36 | 1.14 |
| 13 | 3.24 | 0.17 | 0.64 | 0.34 | 1.28 |
| 14 | 2.63 | 0.16 | 0.62 | 0.32 | 1.24 |
| Mean and standard deviation | | 0.151 ± 0.019 | 0.652 ± 0.088 | 0.301 ± 0.040 | 1.276 ± 0.183 |

**Table 4.** Daily determination of MIC and MBC for the extracts obtained from pill B.

| Day | Released Quantity of Gentamicin (µg/mL) | MIC (µg/mL) | | MBC (µg/mL) | |
|---|---|---|---|---|---|
| | | *P. aeruginosa* | *S. aureus* | *P. aeruginosa* | *S. aureus* |
| 1 | 250.00 | 0.12 | 0.48 | 0.24 | 0.96 |
| 2 | 34.66 | 0.14 | 0.54 | 0.28 | 1.18 |
| 3 | 15.70 | 0.12 | 0.49 | 0.24 | 0.98 |
| 4 | 10.32 | 0.16 | 0.64 | 0.32 | 1.28 |
| 5 | 7.52 | 0.13 | 0.47 | 0.26 | 0.94 |
| 6 | 6.72 | 0.21 | 0.84 | 0.42 | 1.68 |
| 7 | 5.32 | 0.16 | 0.66 | 0.32 | 1.32 |
| 8 | 3.38 | 0.11 | 0.84 | 0.22 | 1.68 |
| 9 | 3.48 | 0.2 | 0.86 | 0.40 | 1.72 |
| 10 | 3.06 | 0.18 | 0.76 | 0.32 | 1.52 |
| 11 | 3.46 | 0.2 | 0.86 | 0.40 | 1.72 |
| 12 | 3.08 | 0.12 | 0.51 | 0.24 | 1.02 |
| 13 | 2.6 | 0.16 | 0.64 | 0.32 | 1.28 |
| 14 | 2.4 | 0.18 | 0.46 | 0.36 | 0.92 |
| Mean and standard deviation | | $0.156 \pm 0.033$ | $0.646 \pm 0.158$ | $0.306 \pm 0.067$ | $1.329 \pm 0.305$ |

## 4. Discussion

Cranial defects may need more than two years for full new bone closure, depending on the size of the reconstruction site [16]. That is why biomaterials such as polymers with a high rate of biodegradation are not suitable as bone graft material in large defects. Of the synthetic polymers and composites used for maxillofacial reconstructions, poly (methyl methacrylate) (PMMA) is widely used [26]. Although it has inert and non-thermoconductive characteristics, it has the disadvantage that if produced on site, it can induce necrosis of the tissues surrounding the implant due to the high polymerization temperature, also leading to the release of toxic monomers [5].

Various combinations of poly(methyl methacrylate) (PMMA), poly(ethylene) (PE), bisphenol A-glycidyl methacrylate (BisGMA) and triethylene glycol dimethacrylate (TEGDMA), hydroxyapatite (HA) and bioactive glasses (BG) were tested and analyzed in order to select the ones with optimal mechanical characteristics, osteoconductive and antibacterial properties [29–31]. Widely used in dentistry for decades and now highly studied as reconstructive materials for craniofacial defects, fiber-reinforced composites show great perspectives in this matter [32–34].

In our previous research, we have developed and characterized E-glass fiber-reinforced composites meant for custom-made craniofacial implants. The developed materials were tested for physical, mechanical and biological properties. The organic matrix showed good conversion rate, with little residual monomer release [2].

The morphological analysis of the external surface of experimental composite materials has shown that the reinforcing material was well-incorporated in the polymer matrix and no monofilaments of fiber glass could be observed on the material surface. The usage of the A 174 silane has allowed the chemical binding of the two phases of the experimental composite materials and created a stable interface, which notably increased the mechanical resistance, e.g., for an FRC with a composition based on Bis-GMA/TEGDMA (60/40)% monomer mixtures and 63% woven fiberglass, the value for flexural strength was 372.45 MPa and for Young's modulus was 13,295 GPa; the copolymer conversion (DC%) after one day was 73.62% [2,22]. The value for the elasticity modulus of the human skull is inscribed in the literature data in the range 1.23–25.83 GPa, with an average of 8.51 GPa and for flexural strength of cranial bones ($82 \pm 25.50$ MPa) [35]. In one study [29], Vallittu reports that after storage in water, there is no significant reduction in flexural strength and modulus of elasticity of FRC due to the hydrolytic effect of water, even after 10 years, due to silane bonds between the glass fibers and the polymer matrix, thus ensuring a stability in time of the FRC. Therefore, the force exerted on the resin matrix is taken over by the woven fiberglass, not allowing the mechanical properties of the material to decrease.

The addition or release of gentamicin have an insignificant effect on the mechanical properties of the material. Although these values may decrease over time, they are much higher than the values obtained by testing the skull bones. These properties are mainly influenced by the degree of glass fiber loading, and respectively by the quality of the adhesion between the resin matrix and the woven fiberglass, as well as how effective the polymerization was.

From a biological point of view, the developed materials proved to be non-cytotoxic (and they even stimulated human dental pulp stem cells and dermal fibroblasts viability in vitro) and produced minor and physiological inflammatory reaction when implanted in subcutaneous and muscle tissues. In this regard, four similar FRC materials were previously tested in vivo and in vitro. The aim of the experiment was to demonstrate the biocompatibility of the implant; as such, initially, the four samples were placed in pulp cells and fibroblast cultures. The results of the study showed no cytotoxicity. When placed subcutaneously on experimental animals, the implants showed good biocompatibility by inducing reduced inflammatory response and homogenous surrounding tissue encapsulation [6]. As for the antimicrobial properties obtained by adding gentamicin on the outer surface of the material, both adherence and bacterial growth on and around the implants were significantly reduced [18].

In order to have predictable and germ-free tissue integration, the bacteria that may contaminate the surface of the implant should be inhibited over a period of two weeks—the time necessary for host cells to cover the implanted material.

In the present study, to improve the antimicrobial effect by controlled releasing of gentamicin over a longer period, the antibiotic was added in all the material's layers (FRC matrix), aiming that its daily concentration around the implant remain in the therapeutic range (MIC) indicated in the literature.

The next step of our study was to find the most suitable solvent in which as much residual monomer would be eluted as possible, so that after its extraction, the implant should have the capacity to continuously and constantly release the amount of gentamicin necessary to prevent the post-surgical infections. Ideally, during the polymerization reaction, the entire monomer content of the resin should be transformed into a polymeric form. Because there is no complete conversion to polymerization, some of the monomers remain unreacted and partially trapped in the polymer matrix. There are numerous studies on the elution of the residual monomer from hardened composites in different mediums. Choosing the right solvent depends on the research purpose. Water and aqueous solutions (artificial saliva, buffers and cell culture media) or organic solvents (ethanol, acetone, chloroform, acetonitrile, tetrahydrofuran) are some of the solvents currently used in the study of residual monomer release. Different opinions regarding the complete elution of the residual monomers can be found in the latest research. Results vary from one to seven days for full release at room temperature. Still, some authors believe that the residual monomer continues to be released after in vivo implantation [36–42].

The efforts of minimizing the amount of residual monomer are justified giving the fact that Bis-GMA and TEGDMA and other methacrylic monomers were proved to present cytotoxicity and genotoxicity [41]. Furthermore, there are studies that emphasize the allergic effect of the methacrylic monomers [43]. It has been shown that the continuous existence of residual monomer after full processed polymerization can tamper with the material proprieties including elasticity, mechanical resistance and lower life span [44,45].

In our study, we tested three organic solvents (75% ethyl alcohol, acetone and chloroform) searching for the most effective solution in terms of complete residual monomer release after the shortest solvent exposure (less than 8 h treatment). The results point out that Bis-GMA based resins stored in acetone or chloroform in order to release the residual monomer show the lowest risk of remaining of unreacted monomer that can inflict host response to implant presence. Comparable results were found in other study [2].

Infections associated with implantation of biomaterials are an unwanted complication of reconstructive surgery, often resulting in prolonged hospitalization, morphological and functional impairment [46]. Perioperative antibiotic prophylaxis leads to a decrease in the rate of infections associated with the implants but does not eradicate them [47–50]. Moreover, the implantation of

biomaterials affects the host's innate local response and may increase the risk of infection [51]. As a result, an antibacterial implant is needed, which can counteract the deficiency of the local immune response induced by the implant itself.

Controlled release of antibiotics is one of the many approaches proposed and tested for antibacterial characteristics of implant materials. Common antibiotics released locally from the implant surface against potential bacterial infections are gentamicin, tobramycin, vancomycin, rifampicin, cefuroxime, and ciprofloxacin [52–54]. Due to bacterial flora changes in the oral and maxillofacial territory, as well as bacterial resistance to antibiotics, currently, the recommended treatments are based on ciprofloxacin and gentamicin. Unlike other studies, gentamicin in our study is introduced into the material but has been shown to be an optimal choice due to its broad antibacterial spectrum, which includes both Gram-positive and Gram-negative bacteria. Moreover, gentamicin has a low rate of allergies, free solubility in water and bacterial resistance to it is rare.

The investigated FRCs had non-degradable crosslinked three-dimensional polymer matrices, the gentamicin release being mainly controlled by diffusion. The amount of the released gentamicin from the FRCs was influenced by the nature and the crosslink density of the network. In the higher crosslinked polymer networks of the FRC samples belonging to series A, the diffusion was smaller compared to the one from FRCs belonging to series B because of a hampered fluid transport into and out of the network. In both cases, the amount of released gentamicin per day decreased with the increase of storage time.

Thus, in the series A of the FRCs, the average of released gentamicin from 1A, 2A, 3A and 4A samples was almost 3 mg in the first day, followed by 2 mg in the second day and 1.37 in the third day. Between the fourth and the ninth day, the amount of released gentamicin was below 1 mg (0.86 mg in the fourth day, 0.52 mg in the sixth day, 0.30 mg in the seventh day and 0.13 mg in the ninth day), reaching a value of 0.025 mg on the eleventh day.

In the FRCs belonging to series B, because of the smaller degree of crosslinking of the matrices in which the fluid transport is facilitated, the average amount of released gentamicin decreased more slowly, being more than 2 mg in the first 3 days (3.15 mg, 2.43 mg and 2.05 mg, respectively), between 1.20 mg on the fourth day and 1.07 mg on the sixth day and between 0.87 mg on the seventh day and 0.16 mg on the tenth day, finally decreasing to 0.045 mg on the eleventh day. However, even in the last day of the study, the amount of gentamicin released was above the value of the minimum inhibitory concentration for both bacterial strains that the experiment was performed on. After the introduction of gentamicin into the entire mass of the FRC samples, the expected amount of gentamicin released was obtained. The results obtained regarding the diffusion of gentamicin from the FRC samples containing gentamicin allowed the validation of the method of quantitative determination of the extracts, namely, the liquid chromatographic method (HPLC) with pre-column derivatization using phenyl isocyanate as the derivatizing agent.

In the case of the A samples, a relatively significant difference is observed between the amount of gentamicin released by the heat-treated sample 1A and the heat-treated sample 2A. The heat-treated 2A sample releases lower amounts of gentamicin than the non-heat-treated 1A sample over the entire study period, with differences greater than 0.85 mg of gentamicin released between the second and tenth days. This behavior can be explained by the fact that by heat treatment, the conversion of double bonds increases, as well as the crosslinking of the polymer, and consequently, a smaller amount of gentamicin can be extracted from the polymer network.

In the case of the B samples, in which half the amount of Bis-GMA was replaced with the linear polymer PMMA, the subsequent crosslinking of the network is not significantly influenced by the postpolymerization heat treatment, and consequently, the difference between the amount of gentamicin extracted in the heat treated sample 2B is almost equal ("similar") to that obtained in the non-heat treated sample 1B for the entire investigated period (starting with the fifth day, the difference being less than 0.01 mg gentamicin).

As the aim of the study was to find a matrix composition that would release as much gentamicin as possible in the first 2 weeks, for samples 3A, 4A, 3B and 4B, the results after the heat treatment were not presented, as the conclusions were the same.

The materials developed and characterized in this study succeed in exhibiting a prolonged antimicrobial effect, while gentamicin is one of the well-tolerated antibiotics in terms of cytotoxicity on many cell lines, including human osteoblasts [55], fibroblasts and keratinocytes [56]. This aspect is critical, taking into account that antibiotic elution from the implant is time-limited, and that by the moment its antimicrobial properties decrease, the implant should be integrated in the surrounding tissues.

## 5. Conclusions

After 12 days, the cumulative amount of released gentamicin was between 7.05–11.38 mg for samples containing dimethacrylate resins (series A) and between 11.21–14.52 mg for samples with PMMA addition (series B).

The results of the microbiological protocol applied in this study suggested that the inclusion of gentamicin in the implant composition induces an antimicrobial effect. Moreover, this effect is maintained over time, protecting against the postsurgical microbial contamination of the implant. Therefore, adding antibiotics in craniofacial implants will reduce the rate of infections in facial reconstructive surgery. Controlling the quantity of residual monomer is essential for the favorable host response to implant placement.

Analyzing the data from the results of the gentamicin release test, extracted residual monomer determination and SEM examination, we came to the conclusion that the most promising composition for gentamicin-releasing implants was found in samples belonging to series B that contained PMMA beside Bis-GMA and TEGDMA dimethacrylates, and the most suitable solvent for the total extraction of the residual monomer was acetone.

**Author Contributions:** Conceptualization, C.A.P. and M.M.; investigation, D.P., M.V., and M.F.; validation, M.A.M.; supervision, M.E.B.; writing—original draft, A.E.M. All authors have read and agreed to the published version of the manuscript.

**Funding:** This research was funded by the Romanian National Authority for Scientific Research, CNDI-UEFISCDI, PN-II, Project PN-II-PT-PCCA-2013-4-0917, 115/2014 and by PHD grant NO 4052 from 1 October 2016 of Iuliu Hatieganu University of Medicine and Pharmacy Cluj-Napoca.

**Acknowledgments:** Special thanks are awarded to Dan Vodnar for the microbiological experiment.

**Conflicts of Interest:** The authors declare no conflict of interest.

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
