# Peer review of "New Antimicrobial Biomaterials for the Reconstruction of Craniofacial Bone Defects"

_coatings, doi:10.3390/coatings10070678_

Round 1

Reviewer 1 Report

Authors described an antibacterial material and they state that could be used for craniofacial surgery. However no prove of that is present in the manuscript either from biological (biocompatibility, efficacy) or material (mechanical properties) points of view and must be added. Many parts are too speculative or without sufficient level of discussion. In particular a better understanding of the drug delivery kinetics and the influence of the parameters changed must be rationalized. Moreover it is not clear what happens to antibacterial properties after the complete release if gentamycin.

Author Response

Point 1. Authors described an antibacterial material and they state that could be used for craniofacial surgery. However no prove of that is present in the manuscript either from biological (biocompatibility, efficacy) or material (mechanical properties) points of view and must be added. Many parts are too speculative or without sufficient level of discussion. In particular a better understanding of the drug delivery kinetics and the influence of the parameters changed must be rationalized. Moreover it is not clear what happens to antibacterial properties after the complete release if gentamycin.

Response 1:

Biocompatibility, efficacy. Four similar FRC materials were previously tested in vivo and in vitro. The aim of the experiment was to demonstrate the biocompatibility of the implant, as such initially, the four samples were placed in pulp cells and fibroblast cultures. The results of the study showed no cytotoxicity. When placed subcutaneously on experimental animals, the implants showed good biocompstibility by induceind reduced inflammatory response and omogen surrounding tissue incapsulation [Ref. 22].

Mechanical properties. In one of our previous studies [2], for an FRC with a composition based on Bis-GMA / TEGDMA (60/40)% monomer mixtures and 63% woven fiberglass, the value for flexural strength was 372.45 MPa and for Young's modulus was 13,295 GPa; the copolymer conversion (DC%) after one day being 73.62%. The value for the modulus of elasticity of the human skull is inscribed in the literature data in the range 1.23–25.83 GPa, with an average of 8.51 GPa and for flexural strength of cranial bones (82 ± 25.50 MPa) [Ref 35].The force exerted on the resin matrix is taken over by the woven fiberglass, not allowing the mechanical properties of the material to decrease. The addition or release of gentamicin have an insignificant effect on the mechanical properties of the material. Although these values may decrease over time, they are much higher than the values obtained by testing the skull bones.

Drug delivery. The investigated FRCs had non-degradable crosslinked three-dimensional polymer matrices, the gentamicin release being mainly controlled by diffusion. The amount of the released gentamicin from the FRCs was influenced by the nature and the crosslink density of the network. In the higher crosslinked polymer networks of the FRC samples belonging to series A the diffusion was smaller compared to the one from FRCs belonging to series B because of a hampered fluid transport into and out of the network. In both cases the amount of released gentamicin per day decreased with the increase of storage time. Thus, in the series A of the FRCs the average of released gentamicin was almost 3 mg in the first day, followed by 2 mg in the second day and 1.37 in the third day. Between the fourth and the ninth day the amount of released gentamicin was below 1 mg (0.86 mg in the fourth day, 0.52 mg in the sixth day, 0.30 mg in the seventh day and 0.13 mg in the ninth day), reaching a value of 0.025 mg on the eleventh day. In the FRCs belonging to series B, because of the smaller degree of crosslinking of the matrices in which the fluid transport is facilitated, the average amount of released gentamicin decreased more slowly, being more than 2 mg in the first 3 days (3.15 mg, 2.43 mg and respectively 2.05 mg), between 1.20 mg on the fourth day and 1.07 mg on the sixth day and between 0.87 mg on the seventh day and 0.16 mg on the tenth day, finally decreasing to 0.045 mg on the eleventh day.

The antibacterial effect of the implant is very important to work in the first weeks, until the surrounding tissues accept it and help it to integrate. During this period it is very important to prevent and eliminate any type of infection. Our study lasted 14 days. We are aware that it is important that these materials be further tested in the future, but the results obtained so far are very promising.

We have improved the discussions in the manuscript by adding our point of view regarding biocompatibility, mechanical properties, respectively the clearer presentation of our previous research.

Reviewer 2 Report

In this manuscript, the authors attempted to evaluate the effect of elution treatment on the TEGDMA/Bis-GMA retention in the FRC, as well as on the gentamicin release from the FRC. After reviewing this article, it is recommended to consider this manuscript for publication in "Coatings" after responding to the following issues:

  1. Would the gentamicin loading and elution process affect the mechanical strength of the composites?
  2. It is not clear that the authors only considered conducting the heat treatment on group-1 but group-3 and -4.
  3. What is the sample size which was used for evaluating the antibacterial activity of the FRC?
  4. Could the TEGDMA/Bis-GMA monomers contribute to the antibacterial activity of FRC? It is highly recommended to provide the antibacterial test by using the extract of FRC without gentamicin.
  5. UV-Vis spectrometer is higher recommended to be used for quantitative analyzing the color changes of resazurin solution after incubation.

Author Response

Point 1. Would the gentamicin loading and elution process affect the mechanical strength of the composites?

Response 1: In one of our previous studies [2], for an FRC with a composition based on Bis-GMA / TEGDMA (60/40)% monomer mixtures and 63% woven fiberglass, the value for flexural strength was 372.45 MPa and for Young's modulus was 13,295 GPa; the copolymer conversion (DC%) after one day being 73.62%. The value for the modulus of elasticity of the human skull is inscribed in the literature data in the range 1.23–25.83 GPa, with an average of 8.51 GPa and for flexural strength of cranial bones (82 ± 25.50 MPa) [Ref 35]. In the reference study [29], Vallittu reports that after storage in water there is no an significant reduction in flexural strength and modulus of elasticity of FRC due to the hydrolytic effect of water, even after 10 years, due to silane bonds between the glass fibers and the polymer matrix thus ensuring a stability in time of the FRC. Therefore, the force exerted on the resin matrix is ​​taken over by the woven fiberglass, not allowing the mechanical properties of the material to decrease. The addition or release of gentamicin have an insignificant effect on the mechanical properties of the material. Although these values ​​may decrease over time, they are much higher than the values ​​obtained by testing the skull bones. These properties are mainly influenced by the degree of glass fiber loading, and respectively by the quality of the adhesion between the resin matrix and the glass fiber fabric and by the way in which the polymerization took place. However, the duration of the study being 14 days, the mechanical properties could not decrease much.

Point 2. It is not clear that the authors only considered conducting the heat treatment on group-1 but group-3 and -4.

Response 2: In the case of samples A, a relatively significant difference is observed between the amount of gentamicin released by the heat-treated sample 1A and the heat-treated sample 2A. The heat-treated 2A sample releases lower amounts of gentamicin than the non-heat-treated 1A sample over the entire study period, with differences greater than 0.85 mg of gentamicin released between the second and tenth days. This behavior can be explained by the fact that by heat treatment, the conversion of double bonds increases, as well as the crosslinking of the polymer, and consequently a smaller amount of getamycin will be able to be extracted from the polymer network. In the case of samples B, in which half the amount of Bis-GMA was replaced with the linear polymer PMMA, the subsequent crosslinking of the network is not significantly influenced by the postpolymerization heat treatment, and consequently the difference between the amount of gentamicin extracted in the heat treated sample 2B is almost equal (“similar”) to that obtained in the non-heat treated sample 1B for the entire investigated period (starting with the fifth day the difference being less than 0.01 mg gentamicin). As the aim of the study was to find a matrix composition that would release as much gentamicin as possible in the first 2 weeks, for samples 3A, 4A and 3B and 4B, respectively, the results after the heat treatment were not presented, because the conclusions being same.

Point 3. What is the sample size which was used for evaluating the antibacterial activity of the FRC?

Response 3: The samples size which were used for evaluating the antibacterial activity of the FRC discs was 6/2 mm.

Point 4. Could the TEGDMA/Bis-GMA monomers contribute to the antibacterial activity of FRC? It is highly recommended to provide the antibacterial test by using the extract of FRC without gentamicin.

Response 4: In one of our previous studies [2] a biological test was performed for an FRC sample with a composition based on Bis-GMA / TEGDMA (60/40)% monomer mixtures and 63% woven fiberglass and the inflammation score was 7.3 ± 0.674. According to ISO 10993-6 recommendations, the FRCs tested were considered mild irritants (average score between 3.0 and 8.9) and no significant difference between groups could be assessed. We believe that some antibacterial activity could be induced additional and of the hydroxyapatite powder from the FRC composition. We are sorry that due to the situation imposed by COVID-19, at the moment, we cannot perform an additional test as you recommended. However, we hope that additional information related to performing the biological test (FRC without gentamicin) will be taken into account.

Point 5. UV-Vis spectrometer is higher recommended to be used for quantitative analyzing the color changes of resazurin solution after incubation.

Response 5: We know that UV-Vis spectrometer is highly recommended  to  analize the color change after resazurin  exposure, but this was our  disponibility and habitude.

Reviewer 3 Report

The manuscript is quite well written. It deals with evaluation of composite material for the medical application i.e. bone defects reconstruction. The authors presents material which shows prolonged antibiotic release and therefore lasting prolonged antimicrobial effect. The presented research can be considered as an interesting development of the research devoted for bone reconstruction and the manuscript is recommended for publication after addressing the following items:

- Authors (even by the article title) suggest that the presented biomaterial is new, but there is a lack of direct and simple explanation what are the differences between this particular composite system and previously published ones (from ref 2, 18 and 22). What is more, if the composition and/or way of preparation is new and material properties are resulting from that (“new antimicrobial biomaterials”) it is mandatory to present detail and comprehensive description of the material preparation. Paragraph 2.1.2 is written too vaguely and imprecisely.  

- in soem places in the text names of chemical compounds are misspelled eg. in lines 48, 89, 337

Author Response

Point 1. Authors (even by the article title) suggest that the presented biomaterial is new, but there is a lack of direct and simple explanation what are the differences between this particular composite system and previously published ones (from ref 2, 18 and 22). What is more, if the composition and/or way of preparation is new and material properties are resulting from that (“new antimicrobial biomaterials”) it is mandatory to present detail and comprehensive description of the material preparation.

Response 1: The combinations and proportions of the final FRC compound were chosen based on the results obtained from previous research, which offered us perspective and guidance in improvement. Several compositions of the impregnation material were tested, different fiberglass fabrics, the number of their layers varied, the amount of gentamicin introduced varied compared to the one used at the beginning of the study and the way of gentamicin deposition was modified. The reference 2 presents and characterized two types of impregnation resin aiming to obtain a mixture with a viscosity suitable for impregnation of woven fiberglass. In reference 18, the material was coated with a layer of PMMA in which it was introduced gentamicin and ZrO2. In reference 22, reinforced impregnation resin with different percentages of woven fiberglass or unidirectional fiberglass was used. The novelty of this study is represented by the composition of FRCs, mainly by the fact that the gentamicin was introduced as a component in the composition of the FRC matrices, not just in the outer layer of the material.

Point 2.  Paragraph 2.1.2 is written too vaguely and imprecisely.

Response 2: I redid the paragraph 2.1.2. and I added (line 82) of the manuscript, the novelty present in the study.

Point 3.  in soem places in the text names of chemical compounds are misspelled eg. in lines 48, 89, 337

Response 3: I made the corrections to the name of the chemical compounds.

Round 2

Reviewer 2 Report

The authors have justified the issues I drew. Despite some of the additional experiments which the reviewer asked can't be accomplished due to the pandemic COVID-19, it may not significantly impact the content of the manuscript. Thus, it is recommended to consider this manuscript for publication in the "Coatings".